# CapTrap-seq: a platform-agnostic and quantitative approach for high-fidelity full-length RNA sequencing

Sílvia Carbonell-Sala [1], Tamara Perteghella[1,2], Julien Lagarde[1,3], Hiromi Nishiyori [4], Emilio Palumbo [1], Carme Arnan[1], Hazuki Takahashi [4], Piero Carninci [4,5], Barbara Uszczynska-Ratajczak [1,6] ✉ & Roderic Guigó [1,2] ✉

Long-read RNA sequencing is essential to produce accurate and exhaustive annotation of eukaryotic genomes. Despite advancements in throughput and accuracy, achieving reliable end-to-end identification of RNA transcripts remains a challenge for long-read sequencing methods. To address this limitation, we develop CapTrap-seq, a cDNA library preparation method, which combines the Cap-trapping strategy with oligo(dT) priming to detect 5′ capped, full-length transcripts. In our study, we evaluate the performance of CapTrap-seq alongside other widely used RNA-seq library preparation protocols in human and mouse tissues, employing both ONT and PacBio sequencing technologies. To explore the quantitative capabilities of CapTrap-seq and its accuracy in reconstructing full-length RNA molecules, we implement a capping strategy for synthetic RNA spike-in sequences that mimics the natural 5′cap formation. Our benchmarks, incorporating the Long-read RNA-seq Genome Annotation Assessment Project (LRGASP) data, demonstrate that CapTrap-seq is a competitive, platform-agnostic RNA library preparation method for generating full-length transcript sequences.

The processing of eukaryotic RNA molecules is essential for their functionality, with capping and polyadenylation playing key roles. Capping entails the addition of a modified guanine nucleotide (7-methylguanosine) to the 5′ end of the RNA molecule, while polyadenylation involves the addition of multiple adenosine residues to the 3′ end[1]. These modifications provide stability, facilitate export, and ensure proper protein-coding capacity and biochemical activity of noncoding RNAs[2,3]. Through alternative splice sites, transcription start sites (TSSs), and transcription termination (TTS) or polyA sites, genes generate a diverse range of protein-coding and noncoding RNA molecules[4]. Furthermore, during annotation, the presence of the cap and the poly(A) tail serves as a sequence tag to evaluate transcript completeness[5].

Understanding the complexity of the transcriptome is crucial for unraveling the principles of gene regulation in contexts like cellular differentiation, organismal development, and disease mechanisms[6–8]. This requires the identification of the complete sequence of the transcripts residing in the cells. However, most current RNA sequencing techniques have limitations that impede the sequence of complete RNA transcripts, from the TSS to the TTS[9]. This is primarily attributed to drawbacks in library preparation methods, particularly those utilizing SMART (switching mechanism at RNA termini) technology[10–12]. These methods have two notable constraints. Firstly, their reliance on Reverse Transcriptase (RT) template switching can lead to the generation of spurious cDNA products, including false splice junctions and

[1]Centre for Genomic Regulation (CRG), the Barcelona Institute of Science and Technology, Barcelona, Catalonia, Spain. [2]Universitat Pompeu Fabra, Barcelona, Catalonia, Spain. [3]Flomics Biotech, SL, Carrer de Roc Boronat 31, 08005 Barcelona, Catalonia, Spain. [4]Laboratory for Transcriptome Technology, RIKEN Center for Integrative Medical Sciences (IMS), Yokohama, Kanagawa, Japan. [5]Human Technopole, Milan, Italy. [6]Department of Computational Biology of Noncoding RNA, Institute of Bioorganic Chemistry, Polish Academy of Sciences, Poznan, Poland. ✉e-mail: barbara.uszczynska@gmail.com; roderic.guigo@crg.cat

transcript chimeras[13–15]. Secondly, none of these methods guarantee the 5′-to-3′ completeness of the sequenced product, resulting in a significant proportion of cDNA 5′ ends falling short of actual TSSs[5,16].

Several custom library preparation methods have emerged to address the issue of incomplete transcript termini[17–20]. However, it should be noted that these approaches are often designed for specific platforms and may require additional sample preparation steps, such as efficient ribodepletion. Consequently, their effectiveness in targeting all transcript types may vary[21,22].

Here, we introduce CapTrap-seq, an open-source, non-proprietary, platform-agnostic method that combines the Cap-trapping strategy[23–26] with oligo(dT) priming to detect 5′capped full-length transcripts. CapTrap-seq is being used to produce the transcriptome data used in the GENCODE project[27,28]. Additionally, we describe a protocol for capping synthetic RNA spike-in sequences, which we use to evaluate CapTrap-seq's gene quantification capabilities.

We benchmark CapTrap-seq and other popular library preparation protocols (TeloPrime, direct RNA, and SMARTer) in human (brain and heart) and mouse (brain) tissues, using both ONT and PacBio sequencing platforms. Together with the evaluation produced by the Long-read RNA-seq Genome Annotation Assessment Project (LRGASP)[29], which we have extended here, we demonstrate that CapTrap-seq is a competitive RNA library preparation protocol to produce full-length transcript sequences. Additionally, we show that CapTrap-seq provides accurate quantitative estimates of transcript and gene abundances, comparable to those obtained through more extensive short-read sequence data.

A Catalan translation of the abstract and a non-specialist summary can be found at https://doi.org/10.5281/zenodo.11488161.

## Results

Here, we begin by presenting an overview of the CapTrap-seq protocol, followed by a benchmark analysis against other popular library preparation protocols using the ONT platform in human and mouse brain samples. Next, we investigate the impact of the sequencing platform on CapTrap-seq by sequencing both the human brain and a human heart libraries using PacBio Sequel I and Sequel II platforms, in addition to ONT. Subsequently, we describe a protocol to cap 5′ends of widely used RNA spike-in sequences, and we use them to evaluate the performance of CapTrap-seq for transcript quantification. Finally, we extend our comparative evaluation of CapTrap-seq using samples generated in the LRGASP benchmark project[29].

### CapTrap-seq for full-length transcript identification

The CapTrap-seq protocol (Fig. 1A) builds upon the previously established Cap-trapping approach[23–26], but with specific optimizations for long-read RNA sequencing. The protocol begins with the enrichment of polyadenylated transcripts using the anchored oligo(dT) method for cDNA synthesis (Anchored dT and PolyA+ in Fig. 1A). After the first-strand synthesis, the initial round of selection for full-length transcripts occurs through the Cap-trapping approach[23] (Cap-trapping in Fig. 1A). Cap-trapping approach was used to address the issue of partial cDNA sequences and to enrich full-length cDNAs. In this process, the 5′ cap of intact RNA molecules is modified with biotin, enabling the capture of full-length capped RNAs using streptavidin. To isolate cDNA sequences that accurately replicate the 5′ end of the original RNA, an RNase treatment step is employed, cleaving the single-stranded RNA region that connects the cDNA. This step also removes ribosomal RNAs that lack a cap in their native state. The sequential double-stranded linker ligation to single-stranded cDNA (sscDNA) is a highly specific reaction that accurately recognizes cap and poly(A) tail structures while safeguarding cDNA molecules against degradation[24]. The sscDNA strand is released and subjected to the second round of full-length molecule selection through a 5′ and 3′ ends dependent linker ligation step, where double-stranded linkers are annealed to the 5′ and

3′ ends of the cDNA molecule (Cap & Poly(A)-dependent Linker Ligation in Fig. 1A). The synthesis of the second strand commences with the ligation of universal adapters to the cDNA molecule flanked by both 5′ and 3′ linkers. By employing universal primers, a Long and Accurate PCR (LA-PCR)[30] method amplifies longer cDNA templates with exceptional fidelity. This approach effectively enriches the presence of full-length cDNA molecules in the resulting libraries to the desired extent (Full-length cDNA library synthesis in Fig. 1A). In summary, the CapTrap-seq protocol utilizes two consecutive rounds of full-length transcripts selection, focusing on the 5′ cap and poly(A) ends, to accurately identify complete cDNA molecules.

### Benchmarking CapTrap-seq and other long-read library preparation protocols

To evaluate the capability of CapTrap-seq in detecting complete and intact transcripts, we conducted a comparative analysis with three popular state-of-the-art library preparation methods (Fig. 1B). These methods included (i) SMARTer cDNA synthesis from Takara Bio[10,31], (ii) the Oxford Nanopore kit for direct RNA sequencing, and (iii) the TeloPrime approach from Lexogen for full-length cDNA amplification[31]. Our study focused on RNA samples derived from the human brain, a tissue known for its high level of transcriptional complexity. For sequencing, we employed the Oxford Nanopore Technologies (ONT) platform, which allows for both cDNA and direct RNA sequencing. We additionally produced short-read RNA-seq data (25 million 125 bp long paired-end reads) using the SMARTer protocol on corresponding brain samples.

All protocols, except direct RNA sequencing, produced high total numbers of long-reads, with high mapping rates, in particular for CapTrap-seq and TeloPrime (>99%). TeloPrime reads mapped to the genome were slightly longer than those produced by CapTrap-seq, which were, in turn, longer than SMARTer, and direct RNA reads (Fig. 1D). CapTrap-seq produced the lowest proportion of sequencing errors compared to the other library preparation methods, while this proportion was the highest for direct RNA sequencing (Supplementary Fig. 1A). Among the tested protocols, CapTrap-seq demonstrated the most uniform read coverage along the length of the GENCODE annotated transcripts (Fig. 1C). The lower performance of direct RNA sequencing suggests that this protocol may be particularly sensitive to the integrity of RNA, which tends to be lower when extracted from the brain (RIN = 6.5 in the case of our sample). To assess this hypothesis, we sequenced RNA extracted from the human heart (RIN = 9.7). Indeed, the number of reads and their length were substantially larger and error rates were lower for heart RNA compared to the brain (Supplementary Fig. 1B, C).

We observed differences among the GENCODE biotypes detected by the different technologies when reads were mapped to annotated genes (GENCODE v24, Supplementary Fig. 1D). The majority of reads for all protocols overlap protein-coding genes, as expected in poly(A) tail-dependent approaches. They also represented a similar proportion of long noncoding RNAs (lncRNAs). However, SMARTer produced a comparatively larger fraction of non-exonic reads, including reads mapping to either intergenic or intronic regions. SMARTer and TeloPrime had a larger fraction of reads mapping to poorly understood miscellaneous RNAs, while CapTrap-seq detected more pseudogenes. Notably, CapTrap-seq almost completely eliminated rRNAs that are not capped in native conditions, thereby minimizing the requirement for an additional ribodepletion step. In terms of nucleotide coverage, approximately 90% of CapTrap-seq's coverage was in genic regions (exons and UTRs), compared to around 60% for TeloPrime and SMARTer (Supplementary Fig. 1E). Additionally, CapTrap-seq produced the lowest percentage of intronic reads (6.6% compared to 23.6% for SMARTer and 17.3% for TeloPrime).

There were also differences in the transcriptional diversity among the protocols, with CapTrap-seq exhibiting a smoother distribution of

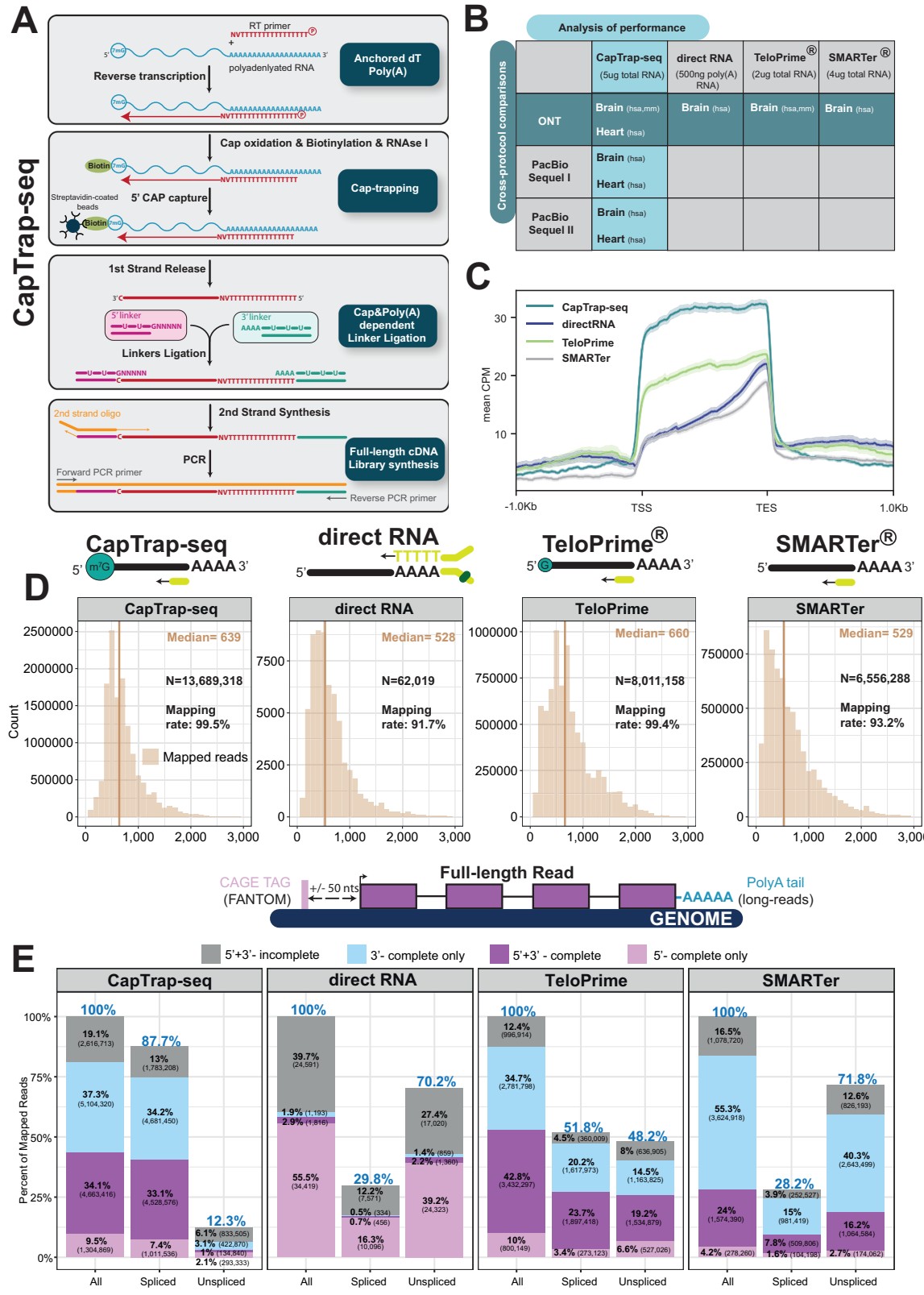

reads across annotated genes. In CapTrap-seq, approximately 10% of the mapped reads aligned to the top ten genes, whereas this proportion was higher (20–25%) for SMARTer and TeloPrime (Supplementary Fig. 2A). As illustrated in Fig. S2C, the transcriptional diversity captured by CapTrap-seq closely resembled that of the Illumina short-read dataset, suggesting that it produces a less biased representation of the transcriptome compared to other protocols.

We assessed the completeness of mapped ONT reads using the presence of unmapped poly(A) tails at the 3′ end and the proximity to CAGE tags[32] (FANTOM5 phase 1 and 2 robust CAGE clusters[33], $N = 201{,}802$) for 5′ cap structures (Fig. 1E). CapTrap-seq and TeloPrime produced a larger proportion of 5′ + 3′ supported reads than SMARTer and direct RNA, with TeloPrime producing overall the highest proportion of 5′ + 3′ supported reads (43% compared to 34% for

**Fig. 1 | Full-length transcript annotation using CapTrap-seq and other library preparation methods. A** CapTrap-seq experimental workflow. Gray boxes highlight the four main steps of full-length (FL) cDNA library construction: Anchored dT Poly(A)+, CAP-trapping[23–26], CAP and Poly(A) dependent linker ligation, and FL-cDNA library enrichment as described in the text. **B** Two adult human complex transcriptomic samples, brain and heart, were used to perform the cross-protocol and cross-platform comparisons to assess the quality of CapTrap-seq. The horizontal green line indicates the cross-protocol comparisons, including four different sequencing library preparation methods: CapTrap-seq, directRNA®, TeloPrime®, and SMARTer®. Whereas, the vertical blue line shows cross-platform comparison using CapTrap-seq in combination with three long-read sequencing platforms: ONT,

PacBio Sequel I, and Sequel II. **C** Read aggregate deepTools2[51] profiles along the body of annotated GENCODE genes. The shaded regions indicate the 95% confidence interval. **D** Length distribution of mapped long-read ONT reads for each protocol. The total number of reads (N), median read length (beige vertical line), and the mapping rate are shown in the top right corner. **E** Detection of full-length reads among all, spliced and unspliced reads, with 5' and 3' termini inferred from robust (FANTOM5 phase 1 and 2 robust ($n = 201,802$)) CAGE clusters[33] and poly(A) tails. Colors highlights four different categories of long-read (LR) completeness: Gray: unsupported LRs; Sky blue: 3' supported LRs; Light pink: 5' supported LRs; Purple: 5' + 3' supported LRs. The blue percentage displayed at the top of each bar indicates the ratio of a specific read type (spliced, unspliced) to the total number of reads.

CapTrap-seq). However, CapTrap-seq exhibited a substantially higher percentage of spliced reads than the other protocols (88% compared to 52% for TeloPrime). Spliced reads are generally more reliable than unspliced reads, as the latter can originate from genomic contamination. Moreover, the proportion of 5' + 3' supported spliced reads was greater for CapTrap-seq compared to the other protocols (33% compared to 24% and 16% for TeloPrime and SMARTer, respectively). CapTrap-Seq, TeloPrime, and Smarter showed constant 5' + 3' support from reads spanning from 600 bp to 3000 bp long (Supplementary Fig. 2B). Shorter reads, as expected, showed reduced end support.

It is important to note that since FANTOM CAGE clusters[32,33] were obtained from samples not included in this study, the presence or absence of CAGE support does not necessarily determine the completeness status of 5' end read. Consequently, the results regarding completeness should be considered indicative rather than definitive. In contrast, our approach to identifying polyadenylated reads has demonstrated reliability, with up to 66% of poly(A) reads supported by the proximity of the canonical polyadenylation motif[34] (Supplementary Fig. 2C). This also validates the authenticity of spliced reads identified by CapTrap-seq, with a support rate of 53%.

The CapTrap-seq efficiency in identifying full-length transcript structures can be illustrated by specific examples. For instance, the *Proline-Rich Mitotic Checkpoint Control Factor* (*PRCC*) gene (genomic and spliced length: ~33 kb and ~2.1 kb, respectively), known for its involvement in pre-mRNA splicing and fusion events with the *Transcription Enhancer Factor 3* (*TEF3*) gene in certain carcinomas, is precisely annotated by CapTrap-seq, capturing the exact TSS, TTS, and exon-intron junctions (Fig. 2A). SMARTer, with its bias towards 3' ends, only detected the 3' terminal exons of *PRCC*, similar to the TeloPrime protocol. CapTrap-Seq can also accurately detect transcript ends for long noncoding RNA genes (lncRNAs). This ability is particularly relevant, as lncRNAs remain the largest, yet the most enigmatic component of our genome[35,36]. In the example in Fig. 2B, both CapTrap-seq and TeloPrime detect transcripts of *MEG3* lncRNA gene, but TeloPrime fails to accurately detect their TSSs. This discrepancy may stem from the fact that TeloPrime was designed to detect G at the 5' end of the transcript rather than the cap structure itself. Nonetheless, in many cases, TeloPrime enables accurate detection of sequenced transcript structures.

Human brain samples provide an ideal setting for stress testing the efficacy of full-length RNA sequencing approaches in real tissues, because of the difficulty of extracting high-quality RNA from this tissue and its great cellular complexity. Therefore, we further tested the CapTrap-seq and the TeloPrime protocols on high-quality adult brain RNA samples from mouse (RIN 9.6), as these may better represent non-human tissue samples.

Like the human brain samples, libraries prepared using these two protocols were sequenced in the ONT platform (Fig. 1B). In mouse, CapTrap-seq results were comparable to those obtained in human, showing consistency in coverage (Supplementary Fig. 3A), read length (Supplementary Fig. 3B), completeness (Supplementary Fig. 3C), and mapping to GENCODE gene biotypes (Supplementary Fig. 3D), while

effectively removing highly abundant, uncapped rRNAs. TeloPrime generated longer reads, and exhibited somewhat superior results in mouse brain compared to human. Overall, both protocols performed similarly in the mouse and human brain samples. However, CapTrap-seq may demonstrate greater resilience to RNA degradation.

## CapTrap-seq performance across sequencing platforms and biological samples

To evaluate the performance of CapTrap-seq across diverse biological samples and sequencing platforms, we employed CapTrap-seq on a human heart sample using ONT, and on both heart and brain samples using two PacBio platforms: PacBio Sequel I (PacBioSI) and PacBio Sequel II (PacBioSII)[37]. In summary, both the human brain and heart libraries were sequenced using ONT, PacBioSI, and PacBioSII (Fig. 1B). The sequencing depth for the heart sample on the ONT platform was comparable to that of the brain sample, while both the brain and heart samples were sequenced to a similar depth on PacBio. However, the number of PacBio reads, particularly PacBioSI, was lower compared to ONT reads (Fig. 3A). Heart reads were, on average, longer than brain reads. The mapping rates and sequencing errors of ONT reads were similar in both heart and brain samples (Supplementary Fig. 4A).

PacBio reads (in the two platforms) were longer than ONT reads, and showed mapping and error rates comparable to ONT reads (Fig. 3A and Supplementary Fig. 4A). The PacBio platforms also demonstrated enhanced detection of polyadenylated reads (Supplementary Fig. 4B). This improvement was accompanied by almost complete validation of poly(A) reads, particularly in the PacBioSII reads, through their proximity to the canonical polyadenylation motif[34] (Supplementary Fig. 4C). As before, we assessed the completeness of the long-read RNA-seq using unmapped poly(A) and CAGE tags. The PacBio platforms produced a larger proportion of full-length and spliced reads than the ONT platform, in particular PacBioSII (Fig. 3B). We did not observe any effect on the read length for reads longer than 400 bp, (Supplementary Fig. 4D).

CapTrap-seq exhibited consistent 5' to 3' uniform read coverage along transcripts across platforms (Supplementary Fig. 5A) and a similar distribution of reads across genes (Supplementary Fig. 5B). Moreover, CapTrap-seq reads mapped to similar gene biotypes and GENCODE annotation (v24), and displayed comparable gene biotypes and successful elimination of rRNA across platforms and samples tested (Supplementary Fig. 5C). Overall, we did not observe any significant difference in CapTrap-seq performance across the human heart and brain samples, except for potential variations attributed to tissue specificity and potentially higher RNA integrity in heart.

The performance of CapTrap-seq in combination with PacBioSII is exemplified by the *HMGCL* (*3-Hydroxy-3-Methylglutaryl-CoA Lyase*) gene. We successfully identified this ubiquitously expressed gene in both heart and brain samples using ONT and PacBioSII (Fig. 3C). However, PacBioSII enabled more accurate detection of this nine-exon gene (>23 kb genomic and ~1.6 kb spliced length). Importantly, the terminal exons, particularly at the 5' end, were accurately identified by all tested platforms.

**A**

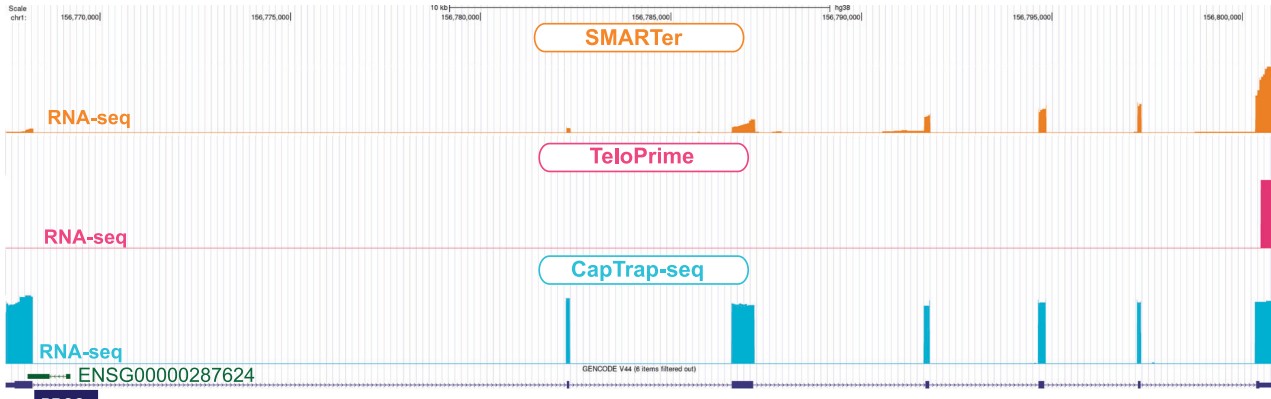

**B**

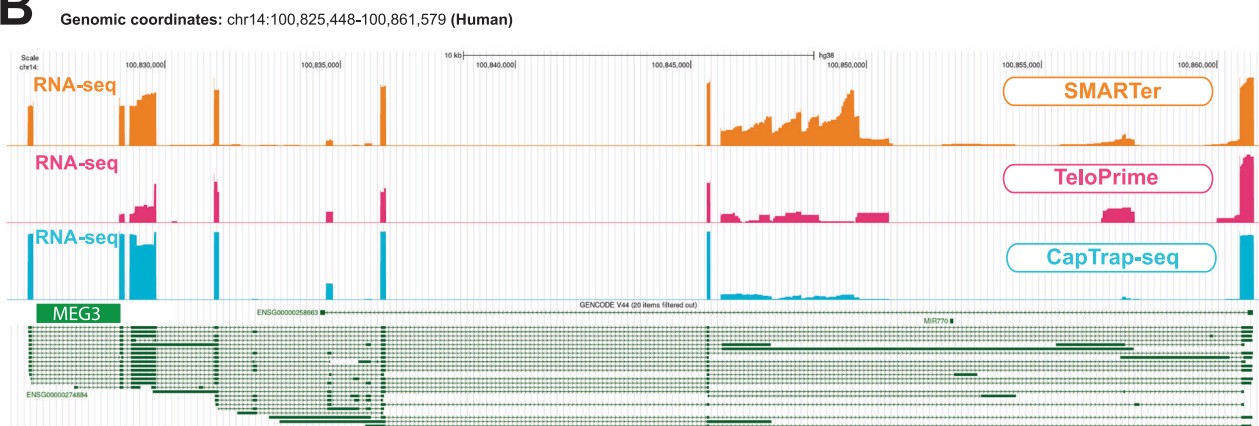

**Fig. 2 | Identification of complete structures of annotated protein-coding and non-protein-coding transcripts in the human genome. A, B** Transcripts were identified for the *PRCC* protein-coding gene (**A**) and for the *MEG3* lncRNA gene (**B**) in human brain samples. Colors denote the library preparation method: orange for SMARTer, pink for TeloPrime, and blue for CapTrap. The GENCODE models (v44) are shown in navy and green for protein-coding and lncRNA genes/transcripts, respectively. The bigwig files derived from the corresponding long-read RNA-seq data were shown below each transcript. All bigwig files are shown using signal tracks displayed in the "full" mode in the UCSC genome browser.

## Capping RNA spike-in controls for reliable full-length transcript detection

Synthetic RNA controls are frequently used to mitigate technical biases in RNA-seq data and analysis, and to calibrate the quantification of transcript abundances. There are two commonly used synthetic RNA spike-in controls: the External RNA Controls Consortium (ERCC) spike-ins with pre-formulated blends of 92 unspliced transcripts to mimic the natural dynamic range of RNA expression, and the Spike-In RNA Variants (SIRVs) designed to capture the transcriptomic complexity with 69 different overlapping isoforms grouped in seven gene modules[38–40]. However, these controls lack a cap structure at their 5′ ends, limiting their compatibility with full-length, cap-dependent transcript sequencing methods like CapTrap-seq.

To overcome this limitation and to assess the detection capacity and quantification accuracy of CapTrap-seq, we have developed a protocol for capping ERCCs and SIRVs (Fig. 4A). This protocol mimics the natural 5′ cap formation process by introducing a 7-methylguanosine (m7G) cap structure to the synthetic RNA controls through a two-step catalytic process. Firstly, the enzyme guanylyl-transferase (GTase) adds GMP derived from GTP to the pp5′N structure located at the 5′ end of the spike-in sequence. Then, the RNA (guanine-

N7) methyltransferase (N7MTase) adds a methyl group to the guanine (derived from the added GMP) at the N7 position.

The CapTrap-seq and TeloPrime libraries generated from the human brain samples (Fig. 1B) included modified spike-ins, comprising both ERCC and SIRVs with newly synthesized 5′ cap structures. For comparison, we also generated additional libraries including the original unmodified spike-ins. There was about a 40-fold increase in the proportion of reads mapping to spike-ins in the sample with capped compared to the sample with uncapped spike-ins (Fig. 4B), demonstrating the utility of capped spike-ins in assessing the efficiency of full-length transcript sequencing protocols.

We employed ERCC spike-ins at known concentrations to evaluate the quantitative capabilities of CapTrap-seq. Our findings revealed that CapTrap-seq could identify molecules at an approximate concentration of $1.05 \times 10^{-2}$ copies per cell (see Materials and Methods for details). Additionally, a clear linear relationship between ERCC concentrations and read counts was observed, providing strong evidence that CapTrap-seq is effective for transcript quantification. TeloPrime behaved similarly (Fig. 4C).

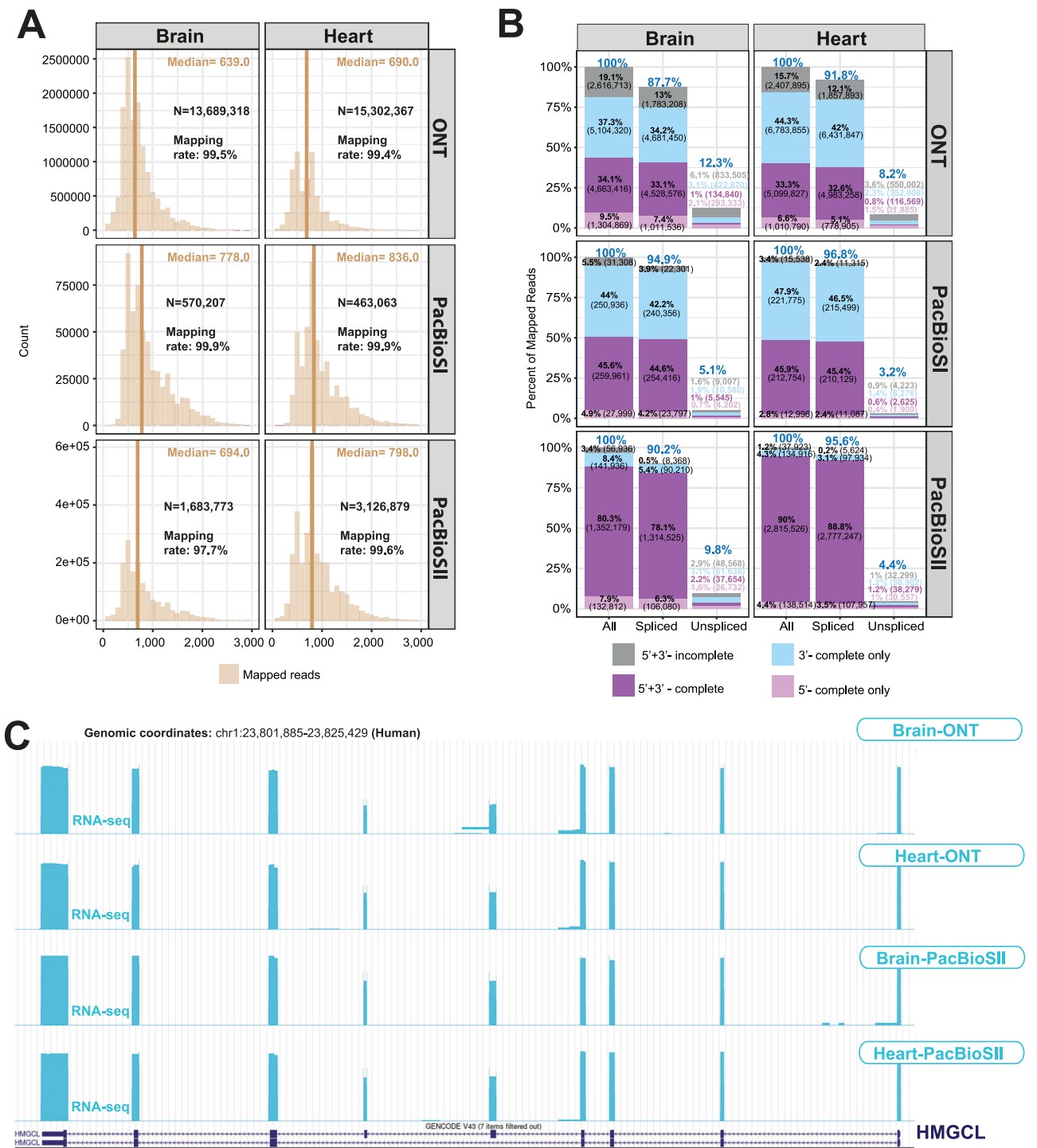

**Fig. 3 | Full-length transcript annotation by CapTrap-seq using different long-read sequencing platforms. A** Length distribution for all mapped reads and **B** the proportion of reads with different types of termini support as described in Fig. 1; **C** CapTrap-seq transcripts for the *HMGCL* gene generated using ONT, the PacBioSI, and SII platforms. For details see Fig. 2.

Subsequently, we employed SIRV spike-ins to assess CapTrap-seq and TeloPrime performance in identifying full-length transcripts. Both protocols demonstrated accurate detection of most SIRVs end-to-end, (Fig. 4D). We also introduced capped spike-ins in the human heart sample that was sequenced using ONT and in the human heart samples sequenced using PacBio (Supplementary Fig. 6A). PacBio showed a similar correlation of expression with ERCCs (Supplementary Fig. 6B), but detected a larger number of SIRVs end-to-end (Supplementary Fig. 6C).

## CapTrap-seq performance in LRGASP

LRGASP, which stands for the Long-read RNA-seq Genome Annotation Assessment Project[29], seeks to evaluate the effectiveness of various experimental and computational protocols in identifying and quantifying transcripts through long-read sequencing technologies, like PacBio and ONT, applicable to both model and non-model organisms. As part of this project, LRGASP generated datasets using diverse platforms and protocols, derived from cDNA and direct RNA samples from human, mouse, and manatee. Here, we specifically focus on

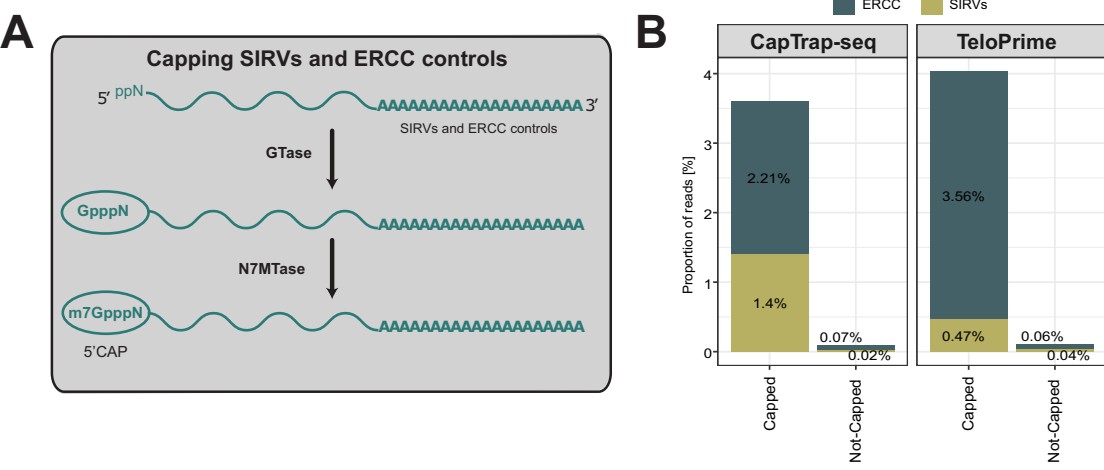

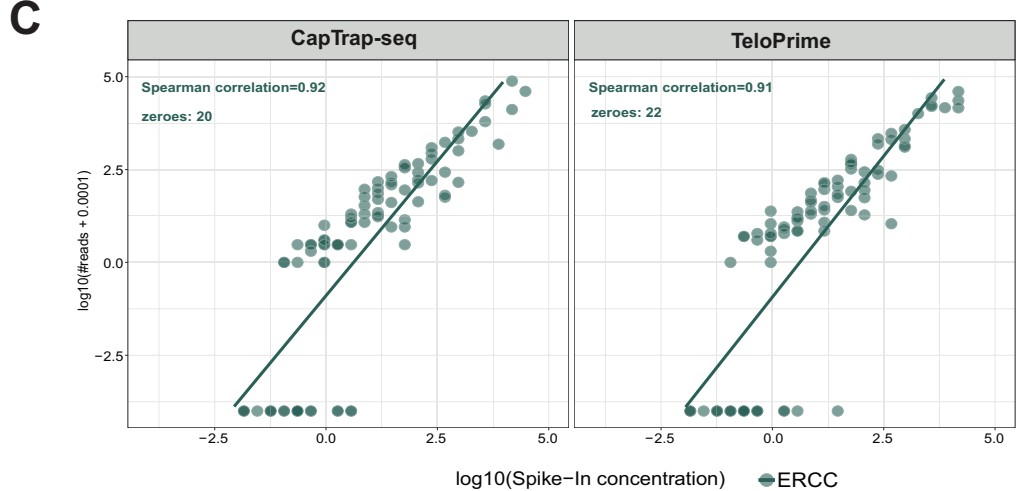

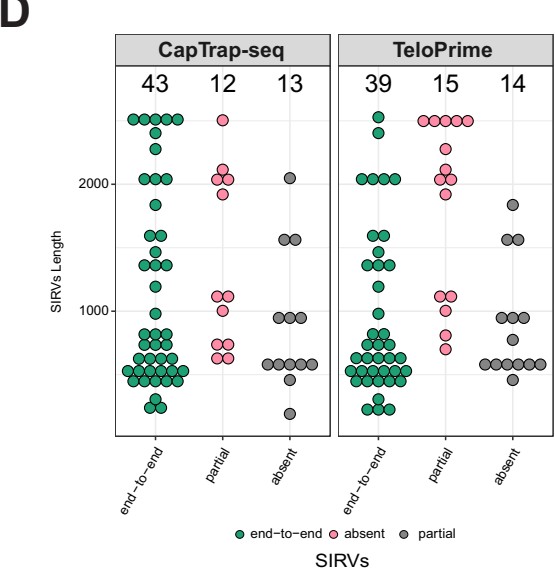

**Fig. 4 | Capping of SIRVs and ERCC controls in the human brain sample using ONT. A** Two-step enzymatic strategy for adding a cap structure at the 5' ends of uncapped RNA spike-in controls; **B** Detection rate of SIRVs (yellow-green) and ERCC (navy) synthetic controls; **C** Correlation between input RNA concentration and raw read counts for ERCC spike-ins in the brain sample. Each point represents a synthetic ERCC control. The green line indicates a linear fit to the corresponding dataset; **D** Detection of SIRVs as a function of length. Three main detection levels have been distinguished: end-to-end (green), partial (red), and not detected/absent (gray). The black numbers displayed at the top indicate the total number of SIRVs for each detection level.

human and mouse samples for which CapTrap-seq libraries were prepared and sequenced.

Four preparation methods were employed to generate libraries sequenced by ONT: CapTrap-seq, direct RNA, direct cDNA, and the R2C2[41] protocol. Additionally, two protocols were sequenced by PacBioSII: CapTrap-seq and SmartSeq2[42]. CapTrap-seq was the exclusive protocol tested on both sequencing platforms (Supplementary Fig. 7A)[29].

Direct comparison between protocols is not straightforward, as some methods explicitly (or implicitly) employ size selection during library preparation. Consequently, reads are considerably longer for R2C2[41] (ONT) and SmartSeq2[42] (PacBio) in contrast to other protocols (Supplementary Fig. 7B). However, R2C2 incurs a cost for its longer and more complete reads. With the same sequencing effort, this protocol yields one to two orders of magnitude fewer reads than other protocols (Supplementary Fig. 7C).

Among the protocols tested with ONT, R2C2 produces a higher proportion of complete transcripts, followed by CapTrap-seq (Supplementary Fig. 7C). In the ones tested with PacBioSII, however, CapTrap-seq yielded similar proportions of complete reads to SmartSeq2, a protocol specifically developed for this platform (Supplementary Fig. 7C).

The LRGASP benchmark already demonstrated the good quantitative performance of CapTrap-seq compared to the other protocols (See Fig. 3B, G in ref. 29). We leveraged the presence of capped RNA spike-ins in the LRGASP samples, generated following the protocol derived here, to further evaluate the sensitivity and quantitative performance of CapTrap-seq, which proves to be the most sensitive among the protocols (Fig. 5A). On the ONT platform, CapTrap-seq typically identifies more than 75% of the ERCC spike-ins, followed by R2C2[41], which typically detects less than 60%. On the PacBio platform, CapTrap-seq is also slightly more sensitive than SmartSeq2[42]. Finally, CapTrap-seq overall stands out as the most quantitative protocol across protocols and platforms, comparable to R2C2 on ONT, and to SmartSeq2 on PacBio (Fig. 5B).

## Discussion

In this study, we introduce CapTrap-seq, a library preparation protocol designed to address the issue of incomplete transcript termini in long-read RNA sequencing methods. CapTrap-seq is an open-source, platform-agnostic, and off-the-shelf approach that aims to produce high-confidence full-length transcript models at high-throughput scale. By filtering out uncapped nucleic acids, it effectively mitigates the risk of genomic DNA and rRNA contamination in both human and mouse samples.

The performance of CapTrap-seq, has been evaluated by the Long-read RNA-seq Genome Annotation Assessment Project (LRGASP) Consortium[29]. The LRGASP analysis revealed that CapTrap-seq enables reliable transcript reconstruction, and produces highly reproducible quantitative estimates of transcript abundances. We leveraged the LRGASP data to complement this benchmark with our own analysis. Our investigation validated the quantitative attributes of CapTrap-seq, and demonstrated its sensitivity in identifying spike-in controls. Furthermore, CapTrap-seq was the sole platform-agnostic protocol in the LRGASP benchmark that successfully operated on both PacBioSII and ONT platforms.

Here, we have further evaluated CapTrap-seq against additional protocols and sequencing platforms. The overall results align with those observed in LRGASP. However, in our samples, CapTrap-seq seems to produce a higher proportion of full-length transcripts on the ONT platform compared to the LRGASP samples. This difference could be attributed to data analysis, notably the use of the previous version of Guppy (v4) by LRGASP for base calling the ONT data.

Overall, these benchmarks indicate that CapTrap-seq stands out as a competitive library preparation protocol, applicable across various platforms for producing 5' complete transcript sequences. CapTrap-seq is actually being used to produce the transcriptome data used in the GENCODE project[27,28], and thousands of CapTrap-seq transcript models have already been included in the GENCODE gene set[28].

In addition to CapTrap-seq, we developed an enzymatic capping strategy for synthetic RNA spike-in controls. This strategy enables efficient capping of the 5' end of both spliced SIRVs and unspliced ERCC controls, regardless of their length and initial concentrations. By employing these capped spike-in synthetic controls, we were able to evaluate the sensitivity, quantitativeness, and accuracy of the CapTrap-seq method. These capped spike-in controls, which mimic the dynamic range of RNA expression and alternative splicing, have been utilized in the LRGASP project, contributing to the precise annotation of full-length transcript models[29].

While CapTrap-seq offers several advantages, it is important to consider its limitations. Firstly, a significant amount of starting RNA (5 μg) is currently required, but efforts are underway to improve the protocol's efficiency with smaller amount of material. Secondly, CapTrap-seq is a relatively complex laboratory procedure, and its multi-stage nature may promote the detection of shorter RNA molecules compared to standard cDNA library preparation methods, such as Switching Template Oligo-based ones. Although no length-specific bias was noted for the ERCC and SIRV controls, which can be up to 2.5 kb long, CapTrap-seq does generate reads that are shorter than those produced by other protocols. Nevertheless, many of these protocols employ some form of (implicit) size-selection step. This feature could also be integrated into CapTrap-seq. Thirdly, CapTrap-seq currently focuses primarily on identifying polyadenylated RNA molecules. Adapting the protocol for the analysis of non-polyadenylated transcriptomic fractions would require substituting the oligo-dT probes and the 3' linker with custom adapters[17,20], although this modification would need further testing.

In conclusion, long-read RNA sequencing, coupled with appropriate data analysis tools, has the potential to produce highly accurate full-length transcripts maps of eukaryotic genomes, without the need for human curation. This is particularly important as projects to sequence the genomes of all eukaryotic species on Earth are underway[43]. We believe CapTrap-seq could play an important role in generating the high-quality transcriptomic data needed to produce accurate annotations of these genomes.

## Methods

### Ethical statement

Given that this study utilizes commercial human RNA samples, ethical approval from an institutional review board was not required. However, all procedures were conducted under ethical principles and guidelines for research involving commercial products. Ethical approval for the use of animals in this study was granted by the PRBB animal facility, from which the animals were purchased.

### Capping ERCC and Lexogen SIRV spike-in controls

The Capping reaction was performed using Vaccinia capping enzyme (catalog num. M2080S, New England BioLabs) following the recommendations of the manufacturer's capping protocol (https://international.neb.com/protocols/0001/01/01/capping-protocol-m2080) with two changes: 3.5 μl of RNAse inhibitors (RNasin Plus RNase Inhibitor, catalog num. N2611, Promega) were added to the capping reaction to avoid RNAse degradation, and the incubation time was extended from 30 min to 2 h, following a recommendation from New England BioLabs

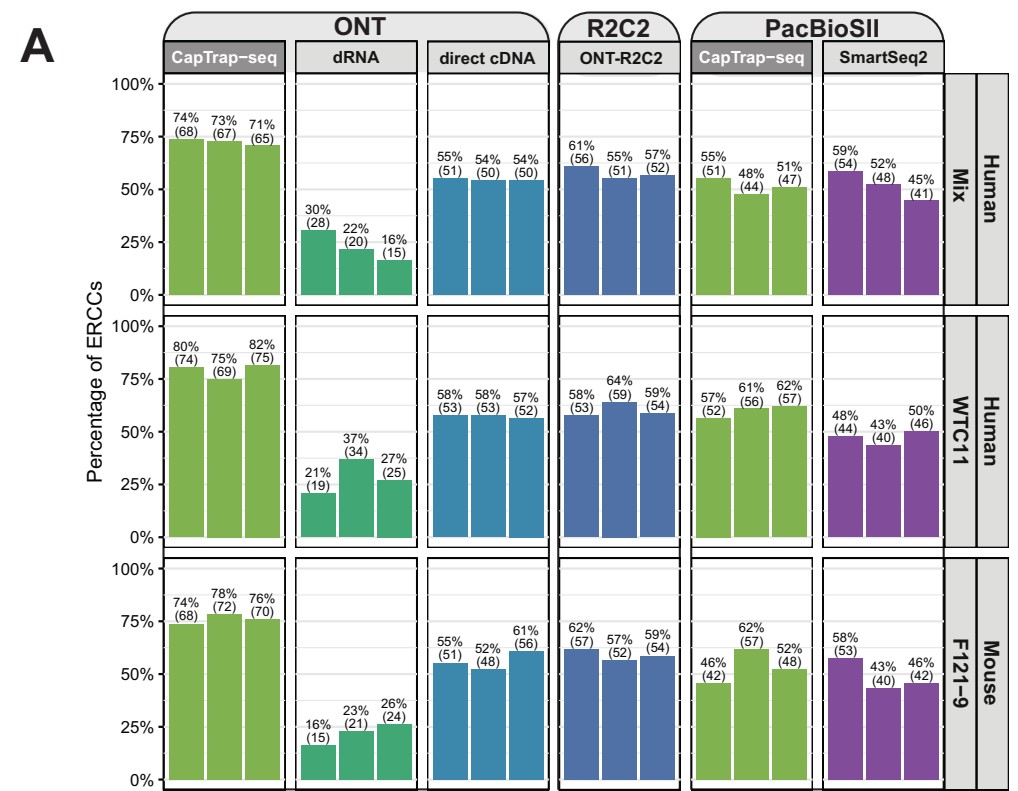

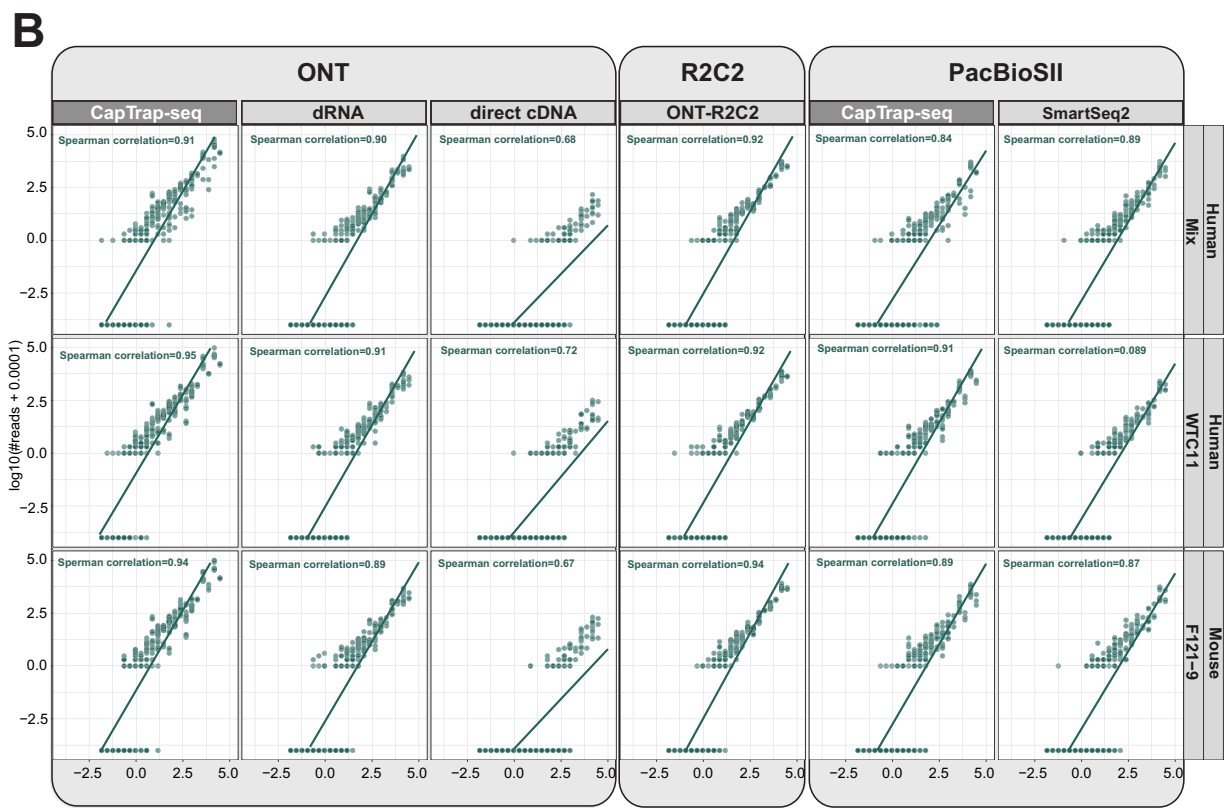

**Fig. 5 | Benchmark of CapTrap-seq using LRGASP samples. A** Proportion of ERCC spike-ins in human and mouse cell lines detected by the different LRGASP protocols and platforms; **B** Correlation between input RNA concentration and raw read counts for ERCC spike-ins. See Fig. 4C for details. For clarity, the triple replicates for each sample have been combined.

technical support scientists. The final capping reaction was purified by using 1.8x AMPure RNA Clean XP beads (catalog num. A63987, Beckman Coulter) and resuspended in 100 μl of nuclease-free water. The full protocol is available at Nature Protocol Exchange[44].

**Samples and cDNA library preparation**
Two total RNA human commercially available heart and brain RNA samples were used to benchmark combinations of library preparation methods and sequencing platforms performance: total RNA from the

adult brain (Ambion brain, catalog num. AM7962; lot num. 1887911 and 1997739, Thermo Fisher) and total RNA from the adult heart (Ambion heart, catalog num. AM7966; lot num. 1866106 and 1906770, Thermo Fisher). Since our study focused on method comparison using commercial samples from Ambion, it did not involve human participants and, therefore, did not require ethical approval. Consequently, factors such as sex and gender did not apply to the study design. The total RNA samples were quality-controlled for concentration and integrity.

Brain total RNA from adult C57BL/6 mice was obtained from the CRG-PRBB animal facility, utilizing flash-frozen tissue samples. RNA extraction was conducted using the PureLink® RNA Mini Kit (catalog num. 12183018A, Life Technologies), following the manufacturer's instructions. Sex and gender were not considered in the study design, as our primary objective was to obtain RNA samples from adult mice solely for protocol comparison. Ethical approval for the study was granted by the PRBB animal facility, from which the animals were purchased.

Quality control assessments for both concentration and integrity were conducted on the total RNA samples obtained from human and mouse before proceeding with the subsequent steps.

The obtained RNA was processed using four different long-read library preparation methods as indicated below:

**CapTrap-seq full-length cDNA libraries (full protocol is available at Nature Protocol Exchange[45]).** The protocol starts with first-strand synthesis where 5 µg of total RNA were mixed in a 10 µl reaction with 1 µl of dNTPs (10 mM; catalog num. 18427013, Life Technologies) and 0.5 µl of following anchored priming oligo: OligodT 16VN (5′ [Phos] TTTTTTTTTTTTTTTTVN 3′, 100 µM, Sigma-Aldrich). The mixture was incubated at 65 °C for 5 min and immediately cooled on ice for 1 min. The enzyme mix (PrimeScript II Reverse Transcriptase, catalog num. 2690A, Takara) was prepared as follows: In a total 10 µl reaction 4 µl of 5x PSII buffer, 1 µl of RNasin Plus RNase Inhibitor (catalog num. N2611, Promega), 4 µl of nuclease-free water (UltraPure DNase/RNase-Free Distilled Water, catalog num. 10977035, Invitrogen) and 1 µl of PrimeScript II RTase (200 U/µl). Ten µl of this enzyme mix were added to each RNA/primer mix reaction in a total volume of 20 µl reaction. First-strand synthesis was performed according to the following program: 42 °C for 60 min and held at 4 °C. The resulting products were purified with 1.8x AMPure RNA Clean XP beads (catalog num. A63987, Beckman Coulter) and resuspended in 42 µl of nuclease-free water.

The purified first strand (40 µl) was then mixed with 2 µl of 1 M NaOAc (pH 4.5) and 2 µl of NaIO₄ (250 mM) in a total volume of 44 µl. The mixture was wrapped with aluminum foil and incubated for 5 min on ice to allow the oxidation of the diol group, present in the m7G cap structure, to aldehyde. Sixteen µl of Tris-HCl (1 M, pH 8.5) were added to the mixture to stop the oxidation reaction, and the whole reaction was quickly purified with 1.8x AMPure RNA Clean XP beads and resuspended in 42 µl of nuclease-free water.

The aldehyde groups resulting in a 40 µl mix were then biotinylated using a mixture containing 4 µl of NaOAc (1 M, pH 6.0) and 4 µl of Biotin (Long Arm) Hydrazide (100 mM, catalog num. SP-1100, Vector Laboratories). The 48 µl resulting mixture was next incubated for 30 min at 40 °C, purified with 1.8x AMPure RNA Clean XP beads, and resuspended in 42 µl of nuclease-free water.

The RNase ONE Ribonuclease (catalog num. M4261, Promega) was used at this point to eliminate the Biotin groups at the 3′ end by degrading only the single-strand RNA present in the mixture. Only the biotin bound to m7G cap structure at 5′ end was kept during this step, where 0.5 µl of RNase ONE enzyme (10 U/µl) and 4.5 µl of 10x RNase ONE Buffer were added to the 40 µl of biotinylated product. The resulting mixture was incubated for 30 min at 37 °C and afterwards, purified with 1.8x AMPure RNA Clean XP beads and resuspended in 42 µl of nuclease-free water.

The m7G cap structure bound to biotin is then selected using M-270 streptavidin magnetic beads. Before starting, 30 µl of M-270

streptavidin magnetic beads (catalog num. 65305, Thermo Fisher Scientific) were washed twice with 30 µl of CapTrap LiCl binding buffer (3.64 ml of nuclease-free water, 35 ml of Lithium chloride 8 M, 0.8 ml of 1 M Tris-HCl at pH 7.5, 0.4 ml of 10% Tween 20 and 0.16 ml of 0.5 M EDTA at pH 8.0) and resuspended with 95 µl of same LiCl binding buffer. The 40 µl of the sample recovered after RNase ONE purification were mixed with 95 µl of washed M-270 streptavidin magnetic beads and incubated at 37 °C for 15 min. During the incubation time, the reaction was mixed by pipetting 10 times at a 7-minute interval to ensure that the beads remained in suspension. Beads were pulled down with a magnet and after supernatant removal, they were washed three times with 150 µl of CapTrap washing buffer (39.12 ml of nuclease-free water, 0.4 ml of 1 M Tris-HCl at pH 7.5, 0.4 ml of 10% Tween 20 and 0.08 ml of 0.5 M EDTA at pH 8.0). After washing steps, the single-strand cDNA was released using 35 µl of CapTrap release buffer (1x RNase ONE buffer with 0.01% Tween 20), heat shocked for 5 minutes at 95 °C, and quickly cooled on ice. The supernatant was then recovered from M-270 streptavidin magnetic beads and stored on ice; meanwhile, a second release was performed, adding 30 µl of CapTrap release buffer, and the supernatant was also collected and mixed with the first release eluate. The final 65 µl volume was treated with 5 µl of an enzymatic mixture containing: 0.1 µl of RNase H (60 U/µl, Ribonuclease H <RNase H>, catalog num. 2150, Takara), 2 µl of RNase ONE (10 U/µl) and 2.9 µl CapTrap release buffer, incubated for 30 minutes at 37 °C and afterward, purified with 1.8x AMPure XP beads (catalog num. A63881, Beckman Coulter) and resuspended in 42 µl of nuclease-free water. The purified sample (approx 40 µl) was concentrated using a speed vac, by drying it for 35 min at 80 °C. The dried sample is resuspended with 4 µl of nuclease-free water.

Two different linkers (custom design, Sigma-Aldrich), a 5′ linker and a 3′ linker, were ligated to the single-stranded cDNA using a 2-step linker ligation[46]. The linkers are double-stranded and have the following sequences:

5′ linker:
GTGGTAUCAACGCAGAGUACGNNNNN-P
P-CACCATAGTTGCGTCTCATG-P
3′ linker:
AAAAAGCAUCGCUGTCTCUTAUACACAUCUCCGAGCCCACGA
GAC-P
CGTAGCGACAGAGAATATGTGTAGAGGCTCGGGTGCTCTG

During the first linker ligation, 1 µl of 5′ linker (10 µM) was ligated with 10 µl of Mighty mix (DNA Ligation Kit <Mighty Mix>, catalog num. 6023, Takara) to 4 µl of the sample coming from the previous step. Just before proceeding with ligation incubation (4 h at 30 °C or 16 h at 16 °C), the linker and the sample are denatured separately for 5 min at 55 and 95 °C, respectively, put on ice for two minutes and then added to the Mighty ligation mix. The 5′ linker ligation product was purified twice, to eliminate the non-incorporated linkers, with 1.8x AMPure XP beads and finally resuspended in 42 µl of nuclease-free water. The resuspended sample (~40 µl) was concentrated using a speed vac, by drying it for 35 min at 80 °C. The dried sample was resuspended with 4 µl of nuclease-free water. During the second linker ligation, 1 µl of 3′ linker (10 µM) was ligated with 10 µl of Mighty mix to 4 µl of the sample coming from the 5′ linker ligation step. Just before proceeding with ligation incubation (4 h at 30 °C or 16 h at 16 °C), the linker and the sample were denatured separately for 5 min at 65 and 95 °C, respectively, put on ice for two minutes and then added to the Mighty ligation mix. The 3′ linker ligation product was purified with 1.8x AMPure XP beads and finally resuspended in 42 µl of nuclease-free water.

The double-stranded linkers were converted into single-stranded to successively allow the second strand synthesis by Shrimp Alkaline Phosphatase (1 U/µl SAP, catalog num. 78390, Affymetrix) and Uracil-Specific Excision Reagent (1 U/µl USER, catalog num. M5505L, NEB) treatment. The 40 µl sample final volume was combined with 2 µl of nuclease-free water, 5 µl of 10x SAP buffer/10xTE, 1 µl of SAP (1 U/µl),

and 2 μl of USER (1 U/μl), mixed, incubated for 30 min at 37 °C, 5 min at 95 °C and finally placed on ice. Afterward, the sample was purified with 1.8x AMPure XP beads and finally resuspended in 42 μl of nuclease-free water.

The samples were concentrated using a speed vac (40 min at 80 °C) and afterward resuspended with 5 μl of nuclease-free water, before second strand synthesis. 20 μl of second strand synthesis mix, prepared with 5.8 μl of nuclease-free water, 12.5 μl of 2x HiFi KAPA mix (catalog num. 7958927001-KK2601, Kapa), 0.5 μl of second primer UMI8 (5′ TCGTCGGCAGCGTCAGATGTGTATAAGAGACAGNNNNNNNNGTGGTA TCAACGCAGAGTAC 3′, 100 μM, custom design, Sigma-Aldrich), containing 8 mer UMI sequence and 1.3 μl of DMSO, were added to 5 μl of the sample. The mixture was incubated for 5 min at 95 °C, 5 min at 55 °C, 30 min at 72 °C and finally held at 4 °C until 1 μl Exonuclease I (20 U/μl, catalog num. M0293S, NEB) was added to each sample. The sample was then incubated for 30 min at 37 °C and afterward, purified twice with 1.8x and 1.4x, respectively AMPure XP beads and finally resuspended in 42 μl of nuclease-free water. The samples were dried up with speed-vac (75 min at 37 °C) and resuspended with 5 μl of nuclease-free water.

These 5 μl were used to amplify the cDNA CapTrap library by Long and Accurate PCR (LA-PCR). In the way to avoid PCR replicates these five μl of each sample were split into two PCR-independent reactions. The PCR reaction mix (TaKaRa LA Taq, catalog num. RR002M, Takara) was assembled using 24 μl of nuclease-free water, 5 μl of 10x Buffer, 5 μl of MgCl$_2$ (25 mM), 8 μl of dNTPs mix (2.5 mM each), 2.5 μl of each primer (forward 5′ TCGTCGGCAGCGTC 3′ and reverse 5′ GTCTCGTGGGCTCGG 3′, Sigma-Aldrich), 0.5 μl of La TaKaRa Taq (5 U/μl) and 2.5 μl of second strand synthesis product, with a final volume of 50 μl. The PCR cycling conditions used were 30 s at 95 °C for the denaturation step, 16 cycles of 15 s at 95 °C, 15 s at 55 °C and 8 min at 65 °C for amplification steps, followed by 10 min at 65 °C and hold at 4 °C. The two PCR replicates were merged and purified with 1x AMPure XP beads and finally resuspended in 22 μl of nuclease-free water. Samples were quantified with Qubit (Qubit 4 Fluorometer, Thermo Fisher Scientific) and quality checked with BioAnalyzer (Agilent 2100 Bioanalyzer, Agilent Technologies).

**SMARTer full-length cDNA libraries.** We used an aliquot of 4 μg total RNA for each sample. Each aliquot of 4 μg was depleted from ribosomal RNA with 10 μl of rRNA removal solution from the Ribo-Zero kit (Ribo-Zero kit, catalog num. MRZH11124, Epicenter-Illlumina), strictly following the manufacturer's protocol. A total of 3.5 μl ribo-depleted material was used for SMARTer (SMARTer PCR cDNA Synthesis Kit, catalog num. 634926 and Advantage 2 PCR kit, catalog num. 639207, Clontech-Takara) protocol. Libraries were prepared strictly following the manufacturer's protocol.

**TeloPrime full-length cDNA libraries.** About 2 μg of each total RNA sample was used to start the TeloPrime cDNA library preparation strictly following the manufacturer's protocol.

**Direct RNA ONT libraries.** The total RNA samples were first poly(A) enriched using Dynabeads® Oligo (dT) following manufacturer protocol. Two rounds of poly(A) enrichment were performed to clean and recover maximum amounts of poly(A) RNA. The full protocol was performed using LiDS as a detergent to prepare washing, lysis, and binding buffers. We repeated the poly(A) enrichment procedure several times, depending on the quality of the RNA sample, until we obtained a total amount of 500 ng of poly(A) RNA. Next, we proceeded with library preparation using the SQK-RNA002 kit. The T4 DNA ligase was used with Quick Ligation Reaction Buffer to anneal and ligate the Direct RNA RT adapters (RTA oligo) to the sample. To create a cDNA-RNA hybrid, adapter-ligated poly(A) RNA was incubated with dNTPs, 5x first-strand buffer, nuclease-free water, and Maxima RT enzyme was used instead of SuperScriptIII to ensemble the reverse transcription

reaction. Tubes were placed in a thermal cycler and incubated for 1 h at 60 °C and 5 min at 85 °C. The reverse-transcription step is highly recommended by the Nanopore protocol to obtain high sequencing throughput of direct RNA samples because it reduces RNA secondary structures and gives stability to the RNA molecule before it passes through the pore and is sequenced. RNA Clean XP beads were used to purify the RT reaction, and finally, the sequencing adapters are ligated to the RNA/cDNA hybrid by using NEBNext Quick Ligation Reaction Buffer and T4 DNA Ligase following the manufacturer's protocol. In the last purification step, RNA Clean XP beads were employed to clean RNA after the adapter-ligation reaction to make it ready for Qubit quantification and fragment distribution quality check.

## ONT and PacBio sequencing

The CapTrap cDNA libraries were split into two different aliquots to meet the sequencing platform requirements. Next, platform-specific sequencing libraries were prepared and loaded into ONT and PacBio long-read sequencing platforms. The TeloPrime, SMARTer, and direct RNA libraries were only sequenced using the ONT platform.

The ONT cDNA sequencing was performed with 500–1000 ng of cDNA sample coming from cDNA-based protocols (CapTrap, SMARTer, and TeloPrime) and strictly following the ONT SQK-LSK108 and SQK-LSK109 adapter-ligation protocols. Direct RNA sequencing was performed with 500 ng of poly(A) RNA. For the ONT sequencing, we used a MinION device, ONT R9.4 flow cells, and the standard Mini-KNOW protocol script.

PacBio Sequel I and Sequel II cDNA sequencing was performed using 500 ng of cDNA samples and following the general protocol workflow for amplicon sample preparation and sequencing SMRTbellTM Express Template Prep Kit 2.0. This workflow allows the generation of highly accurate sequences from amplicons ranging in size from several hundred bases to 10 kb or larger. The samples were prepared following the procedures and details specific to the preparation of cDNA libraries recommended by the manufacturer.

None of the RNA samples and cDNA libraries were subjected to size selection before loading them on ONT and PacBio Sequel (I and II) platforms.

## Illumina short-read SMARTer sequencing

We produced matched short-read RNA-seq data in the human and brain samples. A total of 250 ng of each cDNA (previously prepared for long-read cDNA libraries) were used as starting material. The samples were fragmented using Covaris under the following parameters: 10% Duty cycle, Intensity 5, 200 Cycles per burst, during 30 s in a total volume of 55 μl. Illumina sequencing library was prepared using a TruSeq-based protocol (KAPA LTP Library Preparation kit Illumina, catalog num. 7961898001, Roche) and manufacturer protocol was strictly followed. All samples were barcoded and multiplexed using the Illumina barcoding system (6 mers) and then sequenced in a HiSeq 2500 lane with HiSeq Sequencing v4 Chemistry. A mean of 22.9 million 125-base paired-end reads was generated for each sample. Paired-end reads were then checked for quality and processed together with long-reads.

## Estimating CapTrap-seq sensitivity, quantitativeness, and TM accuracy using spike-ins

The ERCC spike-in mix 1 and Lexogen SIRV_Set 1 mix (generated using 1:1 E1 and E2 mixes to get the maximum dynamic range) using 1:1 proportion ERCCs:SIRVs were added to each sample before the per library preparation. The analysis of the correlation between the initial ERCC spike-in concentrations and raw long-read counts (Fig. 5D and S10C) shows a detection limit of -8.4 × 10$^{-2}$ attomol (−1.875 in log10 units) for sequenced molecules. This threshold is equivalent to 2016 molecules, as 4 μg of a 1:100 dilution of ERCC spike-in was added to 4 μg of each RNA sample. This value approximately equals 1.05 × 10$^{-2}$ molecules per cell on the assumption that the total RNA content of a single cell is 5 pg[47].

To evaluate the accuracy of SIRV transcript reconstruction, we used a complete SIRV annotation containing all 69 SIRV transcripts. The obtained SIRV transcripts were compared against the reference SIRV annotation using gffcompare[48]. The SIRV-Set 1 annotations are available at https://www.lexogen.com/sirvs/download/.

## Sequencing data analysis

The RNA sequencing ONT, PacBio, and Illumina reads were mapped to the human reference genome assembly GRCh38/hg38 (in addition to sequences of 96 ERCC and 69 SIRV spike-in controls). We used Minimap2[49] and STAR (v2.7.6a)[50] to map long and short reads to the genome, respectively. A custom reference gene annotation file was built by combining GENCODE gene annotation (v24), SIRV annotation containing 69 transcripts and 92 ERCC spike-in controls. The read aggregate profiles along the body of annotated GENCODE genes were generated using the deepTools2[51] package.

## Statistics and reproducibility

No statistical methods were used to predetermine the sample size. No data were excluded from the analyses. The experiments were not randomized. The Investigators were not blinded to allocation during experiments and outcome assessment.

## Reporting summary

Further information on research design is available in the Nature Portfolio Reporting Summary linked to this article.

## Data availability

Sequence data, including raw long-read PacBio, ONT and short-read Illumina RNA-seq data have been deposited in the ArrayExpress repository under accession number E-MTAB-13063. The LRGASP data and documentation can be found at https://www.gencodegenes.org/pages/LRGASP/. Source data are provided in this paper and can be accessed at https://github.com/TamaraPerteghella/CapTrap-seq_benchmark_analysis[52].

## Code availability

The code used to process long-read RNA-seq data is available at https://github.com/guigolab/tmerge[53] and https://github.com/julienlag/samToPolyA. Additionally, the custom code for processing the raw data can be found at https://github.com/TamaraPerteghella/CapTrap-seq_benchmark_analysis[52].

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

## Acknowledgements

We thank the Guigó laboratory for their valuable input and help with sample handling. Special thanks are extended to the members of Uszczynska's lab, in particular Monika Kwiatkowska, Marta Blangiewicz, and Tomasz Mądry for their valuable input and assistance in data analysis and figure preparation. We also thank the GENCODE experimental group for their valuable input and discussion, in particular A. Frankish, J.Mudge (EBI, UK) and Rory Johnson (UCD, Ireland), and also thanks to the entire GENCODE group, in particular, M.Diekhans (UCSC, US). We would like to acknowledge the CRG Genomics Unit for assistance with short-read Illumina sequencing and the NGS Sequencing Core facility headed by S. Goodwin for their assistance in generating PacBio Sequel I and Sequel II data. This work and its publication were supported by the National Human Genome Research Institute of the US National Institutes of Health grant (2U24HG007234-09 to R.G.) and the National Science Center PL grant (2018/31/B/NZ2/01940 to B.U.-R.). We acknowledge the support of the Spanish Ministry of Science and Innovation to the EMBL partnership, Centro de Excelencia Severo Ochoa and CERCA Program/Generalitat de Catalunya. We thank R. Garrido Enamorado and R. Carbonell Garcia (CRG), as well as K. Solka and A. Chmielewska (IBCH PAS) for administrative support. Disclaimer: The content is solely the responsibility of the authors and does not necessarily represent the official views of the National Institutes of Health.

## Author contributions

B.U.-R., R.G., S.C.-S., and J.L. designed the experiment. S.C.-S. optimized the CapTrap-seq for long-read sequencing with the input from H.N., H.T., and P.C. S.C.-S. and C.A. generated cDNA/RNA libraries and performed the direct RNA and cDNA sequencing. S.C.-S optimized the capping procedure of RNA spike-ins. J.L. developed the computational pipelines to process the long-read RNA-seq data. B.U.-R., T.P., and E.P. analyzed the data. B.U-R. and R.G. wrote the manuscript with contributions from S.C.-S., T.P., and J.L.

## Competing interests

The authors declare no competing interests.
