## [Peer Review File · Nature Communications]

REVIEWER COMMENTS

Reviewer #1 (Remarks to the Author):

This is a very nice study demonstrating a novel methods for accurately capturing full length transcripts with long read sequencing. This is an important question in the field and the study was nicely done and will provide valuable knowledge to the larger community. I have a few comments:

Can the authors provide any further metric to discriminate between polyadenylated unspliced HCGMs that are true positives vs. genomic DNA contamination risk

While the authors focus on the higher percentage of full-length transcripts, it would be helpful to confirm accuracy of those longer transcripts.

Some additional details regarding the potential missed candidates would be helpful.

Could the authors provide more discussion as to why different platforms capture different transcript subsets independent of sample preparation.

Reviewer #2 (Remarks to the Author):

In the manuscript “CapTrap-Seq: A platform-agnostic and quantitative approach for high-fidelity full-length RNA transcript sequencing” by Carbonell-Sala et al., the authors describe both the CapTrap method for capturing full-length transcripts as well as the Lyric program for the analysis of full-length transcriptome sequencing.

CapTrap is a protocol that aims to generate cDNA covering transcripts end-to-end by using a 5' cap enrichment strategy. Lyric is a computational pipeline which processes CapTrap but also other types of full-length cDNA data.

By combining the evaluation of CapTrap with that of Lyric, the performance of neither is well established.

In order to clearly describe the performance of CapTrap, I suggest analyzing CapTrap data on the level of individual reads - independent of Lyric downstream processing. To establish the performance of Lyric, it should be compared to other isoform identification pipelines.

Major suggestions:

CapTrap

As the manuscript is presented, CapTrap performance, which might in fact be very good, is obscured by Lyric based processing, the comparison to rarely used workflows (TeloPrime and SMARTer), and the analysis of degraded Brain RNA. We suggest the following to address this:

1) All analysis covered in figures 2-4 should be performed on read-level alone, not Lyric-defined transcript models.

2) Because the Brain RNA on which the CapTrap-TeloPrime-SMARTer comparison is based is degraded (RIN <7), High quality LRGASP data including CapTrap but also several other methods of cDNA generation should be analyzed in the same read-level way. This would make it possible to compare CapTrap performance on degraded and intact RNA and to much more widely used full-length transcript sequencing methods.

3) Metrics of completeness should also be stratified for reads of different lengths, maybe in 100nt bins - this could go into the supplement

4) Metrics for completeness should be reexamined. "Spliced TM" and "All TM" categories as shown in 2C and 4A always seem to be having the same number of "5' complete only" and "5' + 3' - incomplete" numbers which is odd.

Lyric

1) The capped SIRVs, sequenced by CapTrap, as introduced in Figure 5 (or generated for LRGASP) can be used to put Lyric in the context of other isoform identification pipelines. The authors erroneously state that Lyric is the only computational tool which doesn't require a genome reference annotation. In fact Bambu, Mandalorion, IsoQuant, and even StringTie2 are documented to run with or without a genome reference. As such, there is no reason to not compare Lyric to these tools. This is especially important since Lyric's performance was somewhat of an outlier in the LRGASP comparison. Comparing Lyric to these 4 tools and establishing Recall and Precision of SIRV isoforms will provide a good idea how the tool performs on CapTrap data and whether the Lyric is indeed required to take full advantage of CapTrap data.

2) Evaluating 3' completeness based on unaligned polyA tails is ill-advised for two reasons. First, The genome contains stretches of As and Ts that these trailing bases can and do align to. This creates False Negatives. Second, and much more frequently, oligo(dT) primers can prime off of A-rich regions of the genome that aren't homopolymers. This will generate polyA tails on reads which will not align to the genome. This creates False Positives. I suggest comparing this approach to experimentally defined polyA sites or polyA signals to establish whether it is reliable.

3) Lyric seems to be similar to SQANTI in evaluating isoforms for completeness based on CAGE peaks. Do these two tools generate the same categorizations? How does Lyric distinguish itself here.

Minor comments:

1) The panels in 4B seem to be doubled.

2) I'm not sure 5C is a good choice to show the underlying data.

REVIEWER COMMENTS

Reviewer #1 (Remarks to the Author):

This is a very nice study demonstrating a novel method for accurately capturing full length transcripts with long read sequencing. This is an important question in the field and the study was nicely done and will provide valuable knowledge to the larger community. I have a few comments:

We appreciate the positive remarks and valuable insights from the Reviewer. In response to the recommendation by the Editor, in this revised version of the manuscript we have focused on CapTrap-seq and we have eliminated any reference to LyRic, which, as a generic tool for long-read RNA-seq analysis, requires its own separate evaluation. Thus, we have not employed transcript models and, in the revised version of the manuscript, we have produced all analyses at the read level. The main results in our manuscript remain essentially intact by the analysis at this level. In addition, although not suggested by the Reviewers, we have generated additional experimental data in order to better understand the behavior of CapTrap-seq and TeloPrime (two additional mouse brain libraries) and of the efficiency of capping of spike ins (two additional human brain libraries). Finally, we have extended our benchmark by leveraging data produced by the LRGASP consortium (Pardo-Palacios, F. J. et al., bioRxiv, 2023). As a result, we believe that this revised version is much improved over our original submission.

1. Can the authors provide any further metric to discriminate between polyadenylated unspliced HCGMs that are true positives vs. genomic DNA contamination risk

We agree with the Reviewer that the possibility of genomic contamination in unspliced RNA-seq derived reads/transcript models remains an important issue in genome annotation. Within GENCODE, for instance, unspliced transcript models are subjected to additional scrutiny and they are often not included in the annotation. Spliced models, in contrast, are unlikely to originate from genomic contamination. We would like to note, in this regard, that CapTrap-seq is among the protocols that produces less unspliced reads. To better visualize this, partially in response to the reviewer, we have modified the layout of figures 1E and 3B.

We also would like to note that, by design, CapTrap-seq and TeloPrime should, in principle, inherently eliminate all molecules lacking the cap structure at the 5' ends. This characteristic is evident in the near-complete removal of uncapped rRNA molecules.

Finally, when investigating this issue, we have observed that the percentages of detected polyadenylation are comparable in spliced and unspliced reads (new Figures S2C and S4C).

While we hope that we have convinced the Reviewer that genomic contamination is not an important issue for CapTrap-seq, we acknowledge that we have not fully responded to their query: how to distinguish genomic contamination from true transcript sequences in RNA-seq reads. One obvious possibility would be to employ the GC content of reads: genomic (non-transcript encoding) reads in the human and mouse genomes should have lower GC content than reads originating from transcript sequences. However, in the case of HCGM reads, these are predicted to be polyadenylated and therefore mapping to the 3' untranslated region of transcripts. While we did observe higher GC content in spliced vs unspliced reads (data not shown), we do not think that

this can be taken as evidence of genomic contamination, as the 3' UTRs have a GC content similar to the human genome background.

2. While the authors focus on the higher percentage of full-length transcripts, it would be helpful to confirm accuracy of those longer transcripts.

In response to the Reviewer, we have generated completeness plots for reads at different lengths (new Figures S2B and S4D). We believe that these are quite interesting, and we thank the Reviewer for suggesting this. End support is lower for shorter reads, as expected, but then from reads in the range from 600 to 3000bp the end support is quite constant for CapTrap-seq, TeloPrime and SMARTer. We have included these results in the manuscript.

3. Some additional details regarding the potential missed candidates would be helpful.

We understand the Reviewer refers to Figures 5C/D and S10B/C in the previous submission of our manuscript (now Figures 4D and S6C), which show that some spike-ins are undetected. This can also be seen in the new Figure 5A, which we have now produced in response to the issue raised by the Reviewer. We would like to note that CapTrap-seq is, among all tested protocols in LRGASP, the one detecting the largest proportion of spike-ins, indicating that it has comparatively high sensitivity.

To better understand the issue raised by the reviewer, we have also modified the layout of new Figures 4D and S6C to specifically show the number of undetected spike ins. We do not observe any effect on the spike-in length in detection rate, neither in CapTrap-seq or TeloPrime (Figures 4D). However, we do observe an impact of the platform, with PacBioSII, in spite of its lower sequencing depth, identifying more end-to-end SIRVs than ONT (Figure S6D). We have included these results in the revised version of the manuscript.

4. Could the authors provide more discussion as to why different platforms capture different transcript subsets independent of sample preparation.

As already pointed out, all the analyses in the revised version of the manuscripts are performed at the read level. Therefore, the revised version excludes the figure that originally illustrated the overlap of transcript models across different platforms.

Nevertheless, the concern brought up by the Reviewer remains valid, and we would like to respond here. We believe that the divergence in transcript subsets captured by ONT and PacBio sequencing arises from significant distinctions in several key aspects. Firstly, both PacBio and Nanopore sequencing technologies exhibit inherent error rates, encompassing base substitutions, insertions, and deletions. These errors are introduced randomly during the sequencing process. Secondly, substantial differences exist in the chemistry and enzymatic processes they entail. PacBio Sequel relies on DNA polymerases that may introduce errors during DNA synthesis, potentially resulting in sequence variations. In contrast, Nanopore sequencing involves the passage of DNA through a protein nanopore, with electrical current measurement as each base traverses the pore. Variations in the speed and position of the DNA strand during translocation through the pore can lead to errors. Thirdly, ONT and PacBio platforms differ in their read-calling approaches. In the case of

ONT, the cDNA molecule undergoes a single pass through the pore, whereas the PacBio strategy employs a rolling circle approach, allowing multiple passes through the same insert and calling the consensus read by merging all the passes. This distinction also influences sequencing depth, which is highest for ONT and considerably lower for the PacBio Sequel platform, with the lowest output for PacBio Sequel I. Lastly, technical considerations inherent to each sequencing platform can bias the sequencing of specific sequences, contributing to variations in the representation of various genomic regions.

Reviewer #2 (Remarks to the Author):

In the manuscript “CapTrap-Seq: A platform-agnostic and quantitative approach for high-fidelity full-length RNA transcript sequencing” by Carbonell-Sala et al., the authors describe both the CapTrap method for capturing full-length transcripts as well as the Lyric program for the analysis of full-length transcriptome sequencing.

CapTrap is a protocol that aims to generate cDNA covering transcripts end-to-end by using a 5' cap enrichment strategy. Lyric is a computational pipeline which processes CapTrap but also other types of full-length cDNA data.

By combining the evaluation of CapTrap with that of Lyric, the performance of neither is well established.

In order to clearly describe the performance of CapTrap, I suggest analyzing CapTrap data on the level of individual reads - independent of Lyric downstream processing. To establish the performance of Lyric, it should be compared to other isoform identification pipelines.

Lyric was initially developed to analyze data produced by CapTrap-seq using the PacBio platform. However, it has evolved to be a generic pipeline to process long-read RNA-seq data. Therefore, the Reviewer is correct that Lyric and CapTrap-seq should be evaluated independently. Following the Editor's strong recommendation to focus our manuscript on the CapTrap-Seq protocol, we have omitted the description of the Lyric pipeline in the revised version. Thus, we have not employed transcript models and, in the revised version of the manuscript, we have produced all analyses at the read level, as suggested by the Reviewer. The main results in our manuscript remain essentially intact by the analysis at this level. In addition, although not suggested by the Reviewers, we have generated additional experimental data in order to better understand the behavior of CapTrap-seq and TeloPrime (two additional mouse brain libraries) and of the efficiency of capping of spike-ins (two additional human brain libraries). Finally, we have extended our benchmark by leveraging data produced by the LRGASP consortium (Pardo-Palacios, F. J. et al., bioRxiv, 2023), as also suggested by the Reviewer.

Major suggestions:

CapTrap

As the manuscript is presented, CapTrap performance, which might in fact be very good, is obscured by Lyric based processing, the comparison to rarely used workflows (TeloPrime and SMARTer), and the analysis of degraded Brain RNA. We suggest the following to address this:

1. All analysis covered in figures 2-4 should be performed on read-level alone, not Lyric-defined transcript models.

In response to the Reviewer's suggestion, we have performed all analyses at the read level (this has affected most of the figures in the paper). We believe that the main messages of the manuscript remain unaltered, but we agree that it provides a less biased evaluation of CapTrap-Seq

2. Because the Brain RNA on which the CapTrap-TeloPrime-SMARTer comparison is based is degraded (RIN <7), High quality LRGASP data including CapTrap but also several other methods of cDNA generation should be analyzed in the same read-level way. This would make it possible to compare CapTrap performance on degraded and intact RNA and to much more widely used full-length transcript sequencing methods.

The selection of the brain as a sample in this study was driven by its transcriptomic complexity and biological relevance, considering it a kind of stress-test for CapTrap-seq. However, following the Reviewer's recommendation, we have expanded our evaluation by producing CapTrap-seq and TeloPrime libraries from less challenging mouse brain RNA and sequenced them using ONT (new Figure S3). The results are overall consistent with those obtained in human.

Additionally, in line with the Reviewer's suggestion, we further expanded our benchmark by incorporating data from the LRGASP study. The LRGASP evaluation had already demonstrated the good quantitative performance of CapTrap-Seq compared to other protocols (See Figures 3B and 3G in Pardo-Palacios, F. J. et al., bioRxiv, 2023). Here, we provide additional benchmarks using the LRGASP data (new Figures 5 and S7).

3. Metrics of completeness should also be stratified for reads of different lengths, maybe in 100nt bins - this could go into the supplement.

In response to this suggestion, we organized the reads by length using 200 nt bins to enhance clarity. Our examination did not reveal any noticeable biases in the proportion of full-length reads across the range of tested lengths for any of the investigated protocols (new Figure S2B) and platforms (new Figure S4D), except for very short reads (0-200 nt), as expected, since they are likely to be incomplete. We also see a drop in support for TeloPrime reads longer than 3000 bps.

4. Metrics for completeness should be reexamined. "Spliced TM" and "All TM" categories as shown in 2C and 4A always seem to be having the same number of "5' complete only" and "5' + 3' - incomplete" numbers which is odd.

We thank the Reviewer for bringing this to our attention. This observation was linked to the requirement in the Lyric analysis for unspliced High Confidence Genome Mappings (HCGMs) to be polyadenylated. As a result, there was no possibility of having unspliced HCGMs/TMs with

complete 5' ends and incomplete 3' ends. However, since the entire analysis has now been conducted at the read level, these results are not included in the revised version of the manuscript.

Lyric

1. The capped SIRVs, sequenced by CapTrap, as introduced in Figure 5 (or generated for LRGASP) can be used to put Lyric in the context of other isoform identification pipelines. The authors erroneously state that Lyric is the only computational tool which doesn't require a genome reference annotation. In fact Bambu, Mandalorion, IsoQuant, and even StringTie2 are documented to run with or without a genome reference. As such, there is no reason to not compare Lyric to these tools. This is especially important since Lyric's performance was somewhat of an outlier in the LRGASP comparison. Comparing Lyric to these 4 tools and establishing Recall and Precision of SIRV isoforms will provide a good idea how the tool performs on CapTrap data and whether the Lyric is indeed required to take full advantage of CapTrap data.

Following the suggestion of the Editor, the revised version of this manuscript focuses exclusively on CapTrap-seq and it does not include Lyric. Therefore, the concerns raised by the reviewer regarding Lyric cannot be addressed in the revised manuscript. Nevertheless, we would like to provide responses to those concerns raised by the Reviewer.

Originally designed as the processing pipeline for CapTrap-seq with the PacBio platform, LyRic has transformed into a fully automated workflow for the analysis of generic long RNA-seq data and gene annotation. LyRic prioritizes the discovery of novel genes in intergenic regions and the identification of missing alternative isoforms of annotated genes. Not relying on pre-existing reference annotations, LyRic employs several filtration steps, including calling High Confidence Genome Mappings (HCGMs) that require the presence of only canonical and high-quality splice junctions for spliced reads, along with the detection of polyA tails for unspliced reads.

We fully acknowledge that other tools can be run both with and without a genome reference, and our earlier statement was a simplification that led to incorrect conclusions. However, LyRic was intentionally designed to operate without genome annotation.

Finally, we would like to emphasize that while Lyric stands out as the most conservative among all the bioinformatics pipelines evaluated by LRGASP, it ranks as the top-performing one in the larger number of metrics employed to evaluate transcript identification, especially when used with the PacBio platform (Figure 2F in the LRGASP manuscript, Pardo-Palacios, F. J. et al., bioRxiv, 2023). The novel transcript models generated by Lyric showed robust support from CAGE and QuantSeq data at their 5' and 3' ends, respectively, as well as from short-read RNASeq data at their splice junctions (Figure 2A in the LRGASP manuscript, Pardo-Palacios, F. J. et al., bioRxiv, 2023). The LRGASP benchmark suggests that Lyric is particularly appropriate for novel transcript discovery from long-read RNA-seq data, aligning with its original design within the GENCODE project.

2. Evaluating 3' completeness based on unaligned polyA tails is ill-advised for two reasons. First, The genome contains stretches of As and Ts that these trailing bases can and do align to. This creates False Negatives. Second, and much more frequently, oligo(dT) primers can

prime off of A-rich regions of the genome that aren't homopolymers. This will generate polyA tails on reads which will not align to the genome. This creates False Positives. I suggest comparing this approach to experimentally defined polyA sites or polyA signals to establish whether it is reliable.

To identify polyadenylated sequencing reads, we employed the in-house developed tool, SamToPolyA (<https://github.com/julienlag/samToPolyA>). SamToPolyA maps polyA sites on the genome based on read mappings in SAM format. It calls a read as poly(A) only when the soft-clipped stretch contains a minimum of ten A or T bases, and the required content of A (or T for the reverse strand) is 80%. SamToPolyA incorporates additional filters to eliminate false positives, including the removal of poly(A) sites, resulting from internal mis-priming during cDNA library construction. Furthermore, we fine-tuned this tool by evaluating the detection of poly(A) sites in synthetic RNA spike-in controls. The suggested comparison confirmed the reliability of this approach, as evidenced by the substantial overlap (60%-99%) between all detected polyA reads and the presence of canonical polyA sites in their proximity. However, as expected we observe a platform specific effect with PacBio platforms generating polyA reads almost fully supported by the canonical polyadenylation motifs. The findings from this analysis are presented in the newly generated figures for cross-protocols (new Figure S2C) and cross-platform comparisons (new Figure S4C). Importantly, the SamToPolyA approach has been incorporated into the LyRic pipeline.

3. Lyric seems to be similar to SQANTI in evaluating isoforms for completeness based on CAGE peaks. Do these two tools generate the same categorizations? How does Lyric distinguish itself here.

LyRic employs a straightforward classification system with four main categories determined by the completeness of 5' and 3' ends, including: (i) full-length transcripts with both 5' and 3' ends complete, (ii) transcripts with a complete 5' end, (iii) transcripts with a complete 3' end, and (iv) fragmentary transcripts. A transcript is considered 5' end complete if it falls within ± 50 bases of the nearest CAGE peak (FANTOM phase 1/2, robust CAGE clusters). Similarly, the 3' end is classified as complete if it contains an unmapped poly(A) tail. In contrast, SQANTI3 classifies isoforms based on the intron chain and uses the transcript termini information to further divide FSM (Full Splice Match) transcripts into subcategories based on their alternative 3' and 5' ends. The described 4-category classification has been employed to categorize the reads for the revised version of the analysis.

Minor comments:

1. The panels in 4B seem to be doubled.

Thank you for pointing this out. The problem has been resolved. The panels that were previously illustrating the read length distribution for all and spliced reads have now been replaced with Figure 3A.

2. I'm not sure 5C is a good choice to show the underlying data.

We agree with this comment. The plots have been substituted with dot plots, a format we believe is more effective in summarizing and visualizing the accuracy of SIRV structure detection. We have also included the total number of SIRVs in each detection category (see new plots 4D, and S6C).

REVIEWERS' COMMENTS

Reviewer #1 (Remarks to the Author):

The authors have addressed my earlier comments.

Reviewer #2 (Remarks to the Author):

I think the authors did a very thorough job replying to my concerns. I did not intend to have the authors drop the evaluation of Lyric entirely, just to evaluate CapTrap-seq and Lyric independently but by doing so, the manuscript is a lot more streamlined.

Evaluating the CapTrap-Seq data on the level of reads makes it much more straightforward to evaluate its performance. As the authors state, CapTrap-seq clearly seems to be a competitive protocol for capturing full-length transcripts. The authors also comprehensively talk about its limitations like slightly shorter reads and high input requirements, thereby painting a complete picture of the method and its use cases.

Thank you for doing such a comprehensive job in reworking the manuscript.